# Current and Innovated Managements for Autoimmune Bullous Skin Disorders: An Overview

**DOI:** 10.3390/jcm11123528

**Published:** 2022-06-19

**Authors:** Kuan-Yu Chu, Hsin-Su Yu, Sebastian Yu

**Affiliations:** 1Department of Dermatology, Kaohsiung Medical University Hospital, Kaohsiung Medical University, Kaohsiung 807378, Taiwan; brandi9304@gmail.com; 2Graduate Institute of Clinical Medicine, College of Medicine, Kaohsiung Medical University, Kaohsiung 807378, Taiwan; 3Department of Dermatology, School of Medicine, College of Medicine, Kaohsiung Medical University, Kaohsiung 807378, Taiwan; 4Neuroscience Research Center, Kaohsiung Medical University, Kaohsiung 807378, Taiwan

**Keywords:** autoimmune bullous dermatoses, pemphigus vulgaris, pemphigus foliaceus, IgA pemphigus, paraneoplastic pemphigus, bullous pemphigoid, epidermolysis bullosa acquisita, mucous membrane pemphigoid, dermatitis herpetiformis

## Abstract

Autoimmune bullous skin disorders are a group of disorders characterized by the formation of numerous blisters and erosions on the skin and/or the mucosal membrane, arising from autoantibodies against the intercellular adhesion molecules and the structural proteins. They can be classified into intraepithelial or subepithelial autoimmune bullous dermatoses based on the location of the targeted antigens. These dermatoses are extremely debilitating and fatal in certain cases, depending on the degree of cutaneous and mucosal involvement. Effective treatments should be implemented promptly. Glucocorticoids serve as the first-line approach due to their rapid onset of therapeutic effects and remission of the acute phase. Nonetheless, long-term applications may lead to major adverse effects that outweigh the benefits. Hence, other adjuvant therapies are mandatory to minimize the potential harm and ameliorate the quality of life. Herein, we summarize the current therapeutic strategies and introduce promising therapies for intractable autoimmune bullous diseases.

## 1. Introduction

Autoimmune bullous skin disorders (AIBDs) are comprised of a constellation of potentially devastating diseases manifested by cutaneous and/or mucosal blisters, evolving eventually into erosions. These clinical features are generated through the binding of immunoglobulin G (IgG) and immunoglobulin A (IgA) against the cutaneous and mucosal adhesion molecules, resulting in detachment of the skin and mucosal layers [1,2]. These autoantigens can locate either on the epidermis or dermal–epidermal basement membrane zone (BMZ) which can thus be categorized into intraepithelial (Table 1) and subepithelial (Table 2) bullous disorders [2].

Direct immunofluorescence (DIF) microscopy of perilesional tissue remains the gold standard of diagnosis for AIBDs. Other assays such as histopathological and serological examinations (e.g., indirect immunofluorescence (IIF) on different substrates, salt-split human skin test, enzyme-linked immunosorbent assay (ELISA) and immunoblotting) further assist the confirmation and can even discern the types of the autoantigens [3,4].

For most AIBDs, the first-line intervention is glucocorticoids (GCs) in combination with immunosuppressive agents [23,24]. Nonetheless, prolonged administration of these conventional therapies brings about numerous adverse effects that overweigh the benefits. Novel approaches have been proposed to maintain therapeutic efficacy and minimize the harm simultaneously. In this review article, we introduce the standard (Table 3) and promising adjuvant therapies established for AIBDs, especially PV and BP.

## 2. Therapeutic Managements

### 2.1. Intraepithelial Autoimmune Bullous Skin Disorders

#### 2.1.1. Pemphigus Vulgaris (PV)

##### First-Line Therapies

*GCs* remain the first-line therapy for PV. Potent topical and intralesional GCs are merely applied to milder cases with limited efficacy [25,26]. Based on the severity of the disease, a loading dose equivalent to 0.5 to 1.5 mg/kg/day of prednisolone is widely accepted. Dosage above 3 to 4 mg/kg/day is not recommended due to the subsequent side effects on the patient [27]. Short-term intravenous pulse therapy (intravenous methylprednisolone 1 g/day or dexamethasone 300 mg/day for 3 consecutive days with an interval of 3–4 weeks followed by 6–8 weeks [28]) can promptly reduce the prednisolone dose and hence cause fewer adverse effects [29,30]. Once through the acute phase, the doctor should taper down the dosage of GCs in order to avoid possible adverse effects such as hypertension, diabetes mellitus, lower resistance to infections, gastrointestinal irritation, osteoporosis, avascular necrosis, glaucoma, and cataracts.

*Rituximab*, a chimeric monoclonal antibody targeting against CD20 antigen on B-cells up to the pre-plasma cell stage, annihilates the autoreactive B-cells in the bloodstream expeditiously and the effects last for at least 6–12 months. It has been regarded as an effective first-line treatment for pemphigus since 2017 [31] and was approved by the U.S. Food and Drug Administration in June 2018 for the treatment of adults with moderate to severe PV. There are currently two strategies for its administration. One is the lymphoma protocol (weekly dosage of 375 mg/m^2^ for four consecutive weeks) and the other is the rheumatoid arthritis protocol (two doses of 1000 mg separated by 2 weeks; may be repeated 6 months later) [32,33,34,35]. Paracetamol and antihistamine are required before each infusion to prevent infusion reactions. In addition, premedication with methylprednisolone 100 mg intravenously is indicated particularly before the first dose of rituximab [36]. Early administration and use in combination with short-term GCs have shown higher efficacy and better therapeutic results. Infusion-related reactions, life-threatening infections, venous thromboembolism, and rarely progressive multifocal leukoencephalopathy are the possible adverse effects [37].

Additional first-line adjuvants including *azathioprine* (1 to 3 mg/kg/day, divided into two separate doses) and *mycophenolate mofetil (MMF)* (2 g/day)/*mycophenolic acid* (1440 mg/day) are administered in combination with GCs rather than as monotherapies due to their late onset. These immunomodulators act as GC-sparing agents with MMF demonstrating less myelosuppressive and hepatotoxic than azathioprine [38].

##### Second-Line Therapies

*Cyclophosphamide (CYP),* given either as 500 mg intravenous bolus or 2 mg/kg/day orally, and *methotrexate (MTX)* 10–20 mg/week, are assumed to be the second-line management in the European Dermatology Forum guidelines [39]. Aggressive hydration or Mesna post CYP is recommended in preventing the development of hemorrhagic cystitis [40]. As for MTX, the adverse effects are dose-dependent and can be mitigated by daily supplement of 1 to 5 mg of folic acid.

*Dapsone* at the dosage of 100 mg/day or up to ≤1.5 mg/kg/day is deemed as the second-line adjuvant treatment [39]. The significance of dapsone in managing PV is controversial. Screening of glucose-6-phosphate dehydrogenase status, complete blood count along with renal and liver functions is widely recommended before commencing dapsone treatment.

*Intravenous immunoglobulin (IVIG)* results in prompt degradation of serum autoantibodies [41]. In each cycle, IVIG is administered via slow infusion at a monthly dosage of 2 g/kg divided into 2 to 5 consecutive days [42]. Nausea, malaise, headache, and fever during infusion are several common, mild and self-limited adverse effects. Despite its effectiveness and safety, its expensiveness and low availability restrict its utilization [43,44].

*Plasmapheresis* non-selectively depletes the plasma proteins along with the pathogenic autoantibodies. Nonetheless, the following rebound phenomenon accelerates the production of the autoantibodies which exacerbates the clinical symptoms [45,46]. GCs and immunosuppressive agents should be given simultaneously to minimize the situation [47,48]. The concomitant complications are sepsis, fluid overload, hyper- or hypotension, hypogammaglobulinemia, hypoproteinemia, and depletion of clotting factors. Another ameliorated apheresis named *immunoadsorption* precisely removes the pathogenic immunoglobulins and preserves fibrinogens, albumins, and other coagulation factors, hence, with fewer adverse effects. Two cycles are performed monthly over 4 consecutive days (2.5-fold plasma volume/day) [49,50,51]. Both plasmapheresis and immunoadsorption are alternative approaches to managing refractory AIBDs that are ineffective or contraindicated to other therapeutic modalities.

##### Emerging Options

Potential new therapeutics of PV are summarized in Figure 1.


**Other anti-CD20 agents: Ofatumumab/Veltuzumab**


Both ofatumumab and veltuzumab are purely humanized anti-CD20 monoclonal antibodies which reduce the possibility of immunological reactions caused by rituximab.

Ofatumumab, the first second-generation agent with high affinity, identifies the extracellular epitope of the CD20 molecule and possesses more potent complement-dependent cytotoxicity than rituximab [52]. Successful treatment following the chronic lymphocytic leukemia protocol (300 mg on day 1, 1000 mg on day 8 with eight concomitant cycles of monthly dosage of 1000 mg) was reported in a case report [53]. Two therapeutic trials in pemphigus were prematurely terminated due to financial issues [54].

Veltuzumab, a type I, second-generation medication with higher binding avidity compared to rituximab can be administered subcutaneously (two doses of 320 mg with an interval of 2 weeks [55]). It was granted Orphan-Drug Designation by the U.S. Food and Drug Administration in 2005 [54].


**Chimeric autoantibody receptor T-cells (CAAR-T-cells)**


CAAR-T-cells are derived from the patient’s autoantibodies engineered to recognize specific antigens located on the target tumor cells and were applied in numerous hematologic malignancies. Ellebrecht et al. manipulated CAAR-T-cells expressing Dsg3-CAAR to combat pathogenic B-cells in vitro [56]. The murine model also showed the Dsg3-CAAR-T-cells eliminate Dsg3-specific B-cells in vivo. A phase 1 trial aiming to establish the maximum tolerated dose of Dsg3-CAAR-T-cells in the mucosal-dominant PV patients is in progress.


**Bruton tyrosine kinase inhibitors: Ibrutinib/Rilzabrutinib (PRN1008)**


Bruton tyrosine kinase (BTK) modulates multiple downstream molecules and is in charge of the survival, proliferation and maturation of the B-cells [57,58]. Through binding with the antigens, Lyn and spleen tyrosine kinase (SYK) activate BTK which then phosphorylates phospholipase Cγ2 (PLCγ2) and initiates multiple inflammatory pathways such as the mitogen-associated protein kinase (MAPK), nuclear factor-κB (NF-κB) and nuclear factor of activated T-cells (NFAT) [59,60]. Ibrutinib was reported to be effective in cases of PNP associated with chronic lymphocytic leukemia [61,62] and mantle cell lymphoma [61,63]. Another oral regimen rilzabrutinib (400–600 mg twice daily for 12 weeks) illustrated an impressive outcome with > 50% of PV patients attaining control of the disease in a phase 2 BELIEVE study [61,64] and it is now undergoing a phase 3 clinical trial in 120 cases of PV and PF [65].


**Anti-B-cell activating factor (BAFF) receptor monoclonal antibody: Ianalumab (VAY736)**


BAFF belongs to the tumor necrosis factor (TNF) family which through binding to its receptor is in charge of the survival of the B-cells [66,67]. A higher serum level of BAFF was noticed in miscellaneous autoimmune diseases, namely rheumatoid arthritis, systemic lupus erythematosus, Sjögren’s syndrome, and systemic sclerosis [68,69,70]. A partial-blind, randomized, placebo-controlled phase 2 clinical trial of ianalumab, a novel and competent IgG1 monoclonal antibody of the BAFF receptor, was applied to 13 PV patients. The efficacy of the drug is not confirmed to date.


**Neonatal Fc receptor (FcRn) antagonists: Efgartigimod (ARGX-113)/SYNT001**


FcRn situated on the endothelial cells of the blood vessel is involved in the homeostasis of IgG and albumin. Once bound to the pathogenic IgGs, it prevents the immunoglobulins from degrading by the lysosomes, thus extending their half-lives [71]. Efgartigimod, an antagonist of FcRn, saturates the receptor, accelerating the degradation of the pathogenic autoantibodies. It demonstrated favorable effects in patients with myasthenia gravis [72] and primary immune thrombocytopenia [73]. Currently, one phase 2 trial in six pemphigus patients is ongoing [74]. Another phase 1/2B clinical trial with SYNT001 recruited eight pemphigus participants but was later terminated due to safety concerns.


**Polyclonal regulatory T-cells (PolyTregs)**


Regulatory T-cells (Tregs) play a significant role upstream in regulating the immune system. Schmidt et al. showed that induction of Tregs downregulates the activation of pathogenic Dsg3-specific T-helper (Th) 2 cells in one HLA-DRB1*04:02 PV transgenic mouse model [75]. Utilization of natural Tregs in lupus, cancer and organ transplantation individuals was proposed [76]. One phase 1 multicenter trial administering intravenous autologous PolyTregs in 12 PV or PF patients is being conducted.

#### 2.1.2. Pemphigus Foliaceus (PF)

The principles in treating PF patients are, in general, identical to those for PV. Associated with lower morbidity and less recalcitrant than PV, the initial management of PF is usually less aggressive [77]. In some mild and localized cases, topical high-potency GCs can achieve satisfactory results.

#### 2.1.3. Pemphigus Erythematosus (PE)

PE is a rare clinical variant of PF [1,78,79]. In mild cases, monotherapy with topical GCs is feasible. For widespread and persistent diseases, systemic prednisolone at the dosage between 0.5 and 1 mg/kg/day in addition to dapsone 100 to 200 mg/day are recommended as the first-line therapies.

#### 2.1.4. IgA Pemphigus

Systemic GCs (0.5 to 1 mg/kg/day prednisolone) and dapsone (100 to 300 mg/day) remain the mainstay therapy for IgA pemphigus. Because HLA-B*13:01 is associated with dapsone hypersensitivity syndrome, genotyping of HLA-B*13:01 before dapsone administration is suggested [80]. In intractable cases, systemic retinoids such as isotretinoin [81] and acitretin [82] can be considered. Howell et al. have proposed a rapid therapeutic response when administering adalimumab (a monoclonal antibody to TNF-α) and MMF to these patients [83].

#### 2.1.5. Paraneoplastic Pemphigus (PNP)

##### Conventional Treatments

Management of PNP is extremely challenging and onerous. Control of the underlying malignancy is the capital issue. Systemic prednisolone (0.5 to 1 mg/kg/day) is prescribed as the first-line treatment [84]. Rituximab, either with the lymphoma or rheumatoid arthritis protocol, is a practical therapeutic approach and has attained successful efficacy in non-Hodgkin lymphoma [85]. Other noteworthy methods include IVIG and immunoadsorption.

##### Emerging Options


**Anti-CD52 agents: Alemtuzumab**


CD52 is a membrane glycoprotein found on numerous immune cells, inclusive of mature lymphocytes [86]. Alemtuzumab, a humanized monoclonal antibody against CD52, causes prompt and long-term depletion of the CD52-bearing lymphocytes [87]. It may be suitable for adjunctive management in PNP cases refractory to other managements, notably in those with hematologic malignancies [88,89,90]. Alemtuzumab is given 30 mg three times per week for 12 weeks along with a daily dose of 40 mg prednisolone.

### 2.2. Subepithelial Autoimmune Bullous Skin Disorders

#### 2.2.1. Bullous Pemphigoid (BP)

##### First-Line Therapies

In BP, stage-adjusted therapy is advocated [28]. According to the involved body surface area (BSA), patients are classified into mild (<10% BSA), moderate (10 to 30% BSA) and severe (>30% BSA) groups. In mild and moderate cases, *topical super-potent GCs* applied to the blisters, erosions and perilesional skin carry out remission. Studies prove that topical clobetasol propionate 40 g/day [91] or 10 to 30 g/day [92] demonstrates equivalent efficacy to systemic GCs and spares their displeasing adverse effects. Nonetheless, the treatment success depends prominently on the adherence and the ability of the patient and the caregiver to apply these topical creams to an extensive body surface area. For rapidly deteriorating and relapsing BPs, *systemic GCs* with initial doses between 0.5 and 0.75 mg/kg/day of prednisolone are suggested [93,94].

Currently, *tetracycline antibiotics* (e.g., doxycycline 200 mg/day) are the most widely endorsed for first-line adjuvant therapy. They can be utilized alone or in combination with *nicotinamide* (up to 2 g/day). Nonetheless, in the real world, relatively short-term effectiveness was reported in one Japanese investigation, revealing that 22 of 27 BP patients managed with doxycycline required second-line prednisolone treatment. [95] Alternative effective regimens such as *azathioprine* (2 to 2.5 mg/kg/day), *MMF* (2 g/day)/*mycophenolic acid* (1440 mg/day), *dapsone* (100 mg/day or up to 1.5 mg/kg/day), and *MTX* (10 to 20 mg/week) have been listed in various guidelines.

##### Second-Line Therapies

For severe and intractable cases, some of the available options include *high-dose IVIG* (2 g/kg per cycle with an interval of 4 to 6 weeks), *immunoadsorption/plasmapheresis*, and *rituximab* (either with the lymphoma or rheumatoid arthritis protocol) [28]. Furthermore, *CYP* given orally with a daily dosage of 2 mg/kg or intravenously with a monthly dose of 15 to 20 mg/kg can be considered.

##### Emerging Options

Potential new therapeutics of BP are summarized in Figure 2.


**Anti-IgE monoclonal antibody: Omalizumab**


Recent research has pronounced that the serum level of IgE-mediated autoantibodies directed against BP180 is correlated to the disease activity of BP [96]. Successful experience of omalizumab, a humanized monoclonal anti-IgE antibody, in BP cases has been broadly reported [97,98,99].


**Anti-C5a receptor (C5aR) antibody: Avdoralimab**


Complement activation, particularly the interaction between the C5a fragment and C5aR, results in mast cell degranulation and the subsequent blister formation in BP [100]. Avdoralimab, a monoclonal antibody against C5aR, is now being evaluated for its efficacy in 40 BP patients in a randomized multicenter phase 2 trial.


**Interleukin-17A (IL-17A) and IL-23 antagonist: Ixekizumab/Ustekinumab/Tildrakizumab**


Higher expression of IL-17 and IL-23 was detected in the serum and the lesional skin of BP patients [101]. Ixekizumab, ustekinumab, and tildrakizumab are all well-known biologics for psoriasis targeting IL-17A, IL-12/IL-23, and IL-23, respectively. As for the BP patients, ixekizumab and ustekinumab are recently undergoing an open-label phase 2 clinical trial, whereas tildrakizumab is in an early phase 1 trial. Further studies are mandatory to support their efficacy.


**Inhibiting the eosinophil cytokines and chemokines: Dupilumab/Bertilimumab/Mepolizumab**


The accumulation of eosinophils in the dermis and the subepidermal clefts is the histological hallmark of BP. With the presence of eosinophils, the elevation of both IL-5 and eotaxin is noted in the blister fluid by the ELISA [102]. Other studies revealed a predominance of Th2 cytokines, inclusive of IL-4, IL-5, and IL-13, in the skin of the BP patients [103,104]. These findings trigger the administration of numerous innovative agents in managing intractable BP cases.

Dupilumab, a monoclonal antibody against IL-4 and IL-13 approved in atopic dermatitis, achieved excellent clearance or satisfactory response in 92.3% of BP patients in a multicenter case series [105]. Bertilimumab is a humanized anti-eotaxin-1 monoclonal antibody. Nine out of eleven BP cases enrolled in a phase 2 clinical trial were prescribed with 10 mg/kg of intravenous bertilimumab on days 0, 14 and 28. Within the follow-up period of 13 weeks, 81% declination of the disease severity was reported without significant adverse effects [106]. Unfortunately, mepolizumab, an IL-5 antagonist sharing a similar mechanism as bertilimumab, failed to show significant differences in the clinical outcome in comparison with the placebo group in a double-blind phase 2 trial. Nonetheless, they still noted a prominent reduction in the serum eosinophil count in the experimental group [107].


**C-C chemokine receptor 3 (CCR3) antagonist: AKST4290**


AKST4290 is an oral antagonist of CCR3, a receptor for eotaxin. Blocking of the CCR3 was proven to cause depletion of eosinophils in animal models [108]. A double-blind, placebo-controlled phase 2 study for AKST4290 has been completed. The participants received AKST4290 at the dosage of 400 mg twice concurrently with mometasone furoate until the disease was under control. No results are available to date.

#### 2.2.2. Mucous Membrane Pemphigoid (MMP)

The therapeutic strategies depend on the afflicted mucosal areas, disease severity, and its progression. Mild to moderate oral MMP can be alleviated by topical GCs [109] and tacrolimus [110]. Severe cases or those with ocular involvement require systemic GCs, dapsone, or immunomodulators for disease control. IVIG, steroid/cyclophosphamide pulse therapy, plasmapheresis, and rituximab are reserved for recalcitrant patients [111].

#### 2.2.3. Linear IgA Bullous Dermatosis

Dapsone contributes to prompt remission and qualifies as the first-line management in linear IgA bullous dermatosis [112]. Topical potent GCs can be applied to cutaneous lesions as the initial treatment. Systemic oral prednisolone 0.25 to 0.5 mg/kg/day should be considered for those who fail to improve from the topical GCs. Additional therapeutic options include sulfonamides, colchicine, and tetracycline/niacinamide.

#### 2.2.4. Epidermolysis Bullosa Acquisita (EBA)

##### Conventional Treatments

Management of EBA is exceptionally arduous. High-dose systemic GCs ranging from 1 to 1.5 mg/kg/day remain the preferred regimen. Dapsone or colchicine can be applied together with GCs to accelerate remission. The administration of other adjuvant therapies such as immunosuppressive agents (e.g., cyclosporine, azathioprine, CYP, MTX, and MMF), IVIG, and rituximab is utilized in the refractory cases [113,114,115,116].

##### Emerging Options


**Anti-CD25 monoclonal antibody: Daclizumab**


The production of autoantibodies in EBA patients is highly associated with T-cells. CD25, a significant component of the IL-2 receptor, regulates the survival and activation of T-cells. Daclizumab, through blocking CD25, facilitates a rapid and continuous reduction in lymphocyte CD25 expression and was proven to be effective in one EBA patient [117].

#### 2.2.5. Dermatitis Herpetiformis (DH)

A strict gluten-free diet takes a chief role in dealing with DH. Slow titration of dapsone is the most popular first-line therapy showing immediate clinical improvement. Pharmacotherapy with sulfa-based regimens (sulfapyridine, sulfasalazine and sulfamethoxypyridazine) is suitable for those who cannot tolerate prior treatments [118,119,120]. Topical GCs may soothe the pruritus; nevertheless, systemic GCs are not warranted in DH.

#### 2.2.6. Laminin γ1 Pemphigoid

The therapeutic information is sparse due to its rarity. Topical GCs can be applied to those with milder symptoms. With higher disease severity, some of the popular selections include systemic GCs (starting from 0.5 mg/kg/day) in collaboration with dapsone (1–1.5 mg/kg/day), cyclosporine, or azathioprine [121,122]. Other documented managements include doxycycline, high-dose IVIG, colchicine, and ustekinumab [122].

## 3. Conclusions

AIBDs consist of a dozen of debilitating diseases that deserve our attention. In recent years, a growing number of investigations have provided us with some novel thoughts for confronting these diseases. In PV, clinical trials targeting autoreactive B-cells including anti-CD20 agents, CAAR-T-cells, BTK inhibitors, and anti-BAFF receptor antibodies are burgeoning fields of research. The same concern has been raised regarding FcRn antagonists and PolyTregs, both of which demonstrate the capabilities of suppressing pathogenic autoantibodies. On the other hand, in BP, the Th2 axis such as mast cells, eosinophils, and IgE is of paramount interest. Notably, biologics against the IL-17/IL-23 inflammatory pathway may be revolutionary choices for recalcitrant cases. In the future, these personalized approaches would usher in a brand new era in managing these diseases.

## Figures and Tables

**Figure 1 jcm-11-03528-f001:**
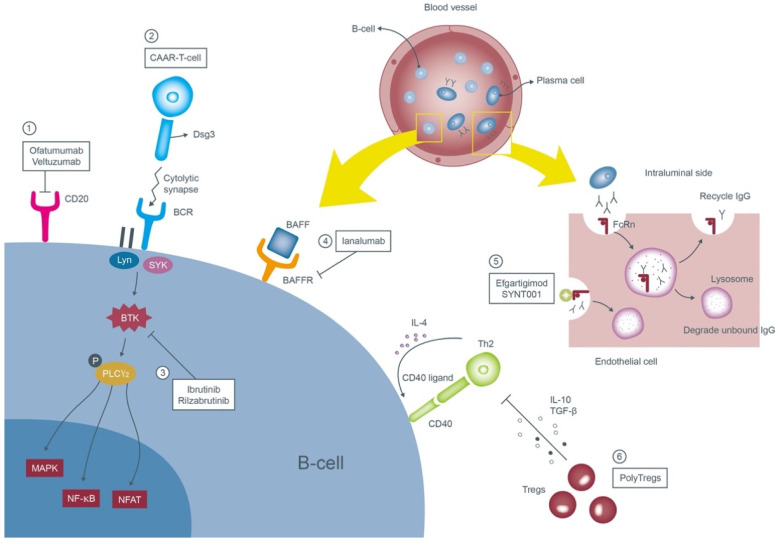
Mechanisms of the emerging therapies of pemphigus vulgaris. These therapies target anti-Dsg3 B-cells or autoantibodies. ① Ofatumumab and veltuzumab are new second generation monoclonal antibodies targeting CD20 on the surface of B-cells. ② Engineered CAAR-T-cell expressing the Dsg3 ectodomain recognizes and forms cytolytic synapse with the pathognomonic B-cells and subsequently annihilates them. ③ Ibrutinib and rilzabrutinib prohibit the proliferation of B-cells through blocking of the BTK. ④ Ianalumab inhibits the signal transduction of BAFF by binding to its receptor and contributes to depletion of B-cells. ⑤ Efgartigimod and SYNT001 occupy the binding sites of anti-Dsg3 antibodies to the FcRn and accelerate their clearance. ⑥ PolyTregs suppress the adaptive immune cells via inhibitory cytokines and terminate the differentiation of B-cells toward plasma cells. Abbreviations: Dsg3, desmoglein 3; CAAR-T-cell, chimeric autoantibody receptor T-cells; BTK, Bruton tyrosine kinase; BAFF, B-cell activating factor; BAFFR, B-cell activating factor receptor; FcRn, neonatal Fc receptor; PolyTregs, polyclonal regulatory T-cells.

**Figure 2 jcm-11-03528-f002:**
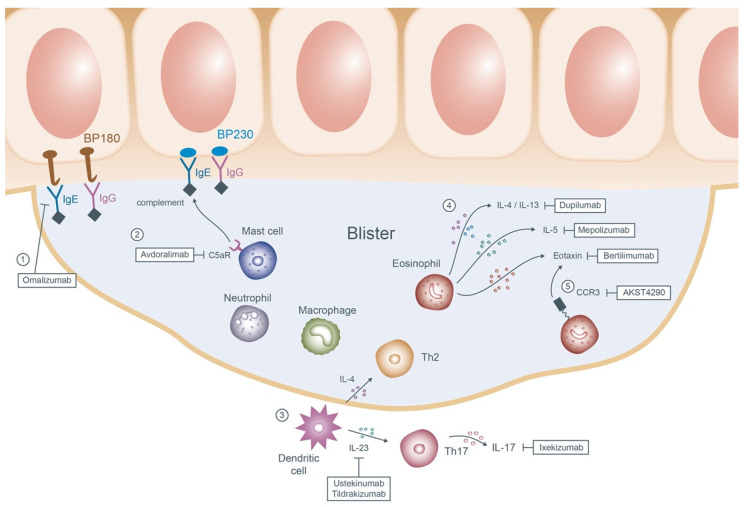
Mechanisms of the emerging therapies of bullous pemphigoid. ① Omalizumab prohibits the adherence of IgE antibodies to the basement membrane proteins BP180 and BP230. ② Avdoralimab blocks the subsequent binding of the C5aR on the mast cells to the complements and deters the process of degranulation. ③ In the upstream, dendritic cells release IL-4 and IL-23 which then activate Th2 and Th17 cells, respectively. Ustekinumab, tildrakizumab and ixekizumab are therefore applied to inhibit the cascade. ④ In the blisters, chemokines result in recruitment of multiple innate and adaptive cells of which eosinophils play the pivotal role. Dupilumab, mepolizumab, and bertilimumab targeting the downstream products of eosinophils (IL-4/IL-13, IL-5 and eotaxin) may reduce further blister formation. ⑤ AKST4290 interacts with the receptor of eotaxin CCR3 causing downregulation of the eosinophils. Abbreviations: IgE, immunoglobulin E; IgG, immunoglobulin G; C5aR, C5a receptor; IL, interleukin; Th, T-helper; CCR3, C-C chemokine receptor 3.

**Table 1 jcm-11-03528-t001:** Clinical characteristics, target autoantigens and diagnosis of the intraepithelial autoimmune bullous skin disorders (AIBDs).

Diseases	Clinical Features	Autoantigens [3]	Diagnosis [3,4]
Pemphigus vulgaris (PV)	Flaccid blisters and painful erosions on skin with propensity to strained and intertriginous areas [5,6] Extensive oral mucosal involvement precedes the skin lesions	Dsg1,3	H&E: Suprabasal acantholysis, tombstone pattern of the basal keratinocytes DIF: Intercellular deposition of IgG/C3 IIF: Intercellular deposition of IgG ELISA: Anti-Dsg3 and/or anti-Dsg1 antibodies
Pemphigus foliaceus (PF)	Flaccid blisters and erosions affect exclusively the cornified skin and spared the mucosal regions Usually in a seborrheic distribution	Dsg1	H&E: Acantholysis and spongiosis within the stratum granulosum DIF: Intercellular deposition of IgG/C3 IIF: Intercellular deposition of IgG ELISA: Anti-Dsg1 antibodies
Pemphigus erythematosus (PE)	Blisters on the erythematous plaques at the nose, nasolabial folds and malar regions [7]	Dsg1	H&E: The same as PF DIF: Intercellular and shaggy basement membrane deposition of IgG/C3 IIF: Intercellular deposition of IgG ELISA: Anti-Dsg1 antibodies Positive serum antinuclear antibody
IgA pemphigus	Vesiculopustular lesions on the erythematous plaques in an annular morphology at the trunk and proximal extremities [8]	Dsc1,2,3 & Dsg1,3	H&E: Prominent intraepidermal neutrophilic infiltrates DIF: Intercellular deposition of IgA/C3 IIF: Intercellular deposition of IgA ELISA: Anti-Dsc1,2,3 and anti-Dsg1,3 IgA antibodies
Paraneoplastic pemphigus (PNP)	Blisters, erosions or lichenoid lesions on the skin Extensive and intractable stomatitis Related to thymoma [9] and hematological malignancies [10,11,12]	Envoplakin, periplakin, BP230, Dsg1,3, desmoplakins, epiplakin, α-2- macroglobulin- like antigen-1, and plectin	H&E: Overt interface/lichenoid infiltrates with dyskeratotic cells, foci suprabasal acantholysis DIF: Intercellular and basement membrane deposition of IgG and/or C3 IIF: Intercellular deposition of IgG (monkey esophagus and monkey/rat bladder [13]) ^1^ ELISA: Common anti-Dsg3 and anti-envoplakin antibodies

Abbreviations: H&E, hematoxylin and eosin stain. DIF/IIF, direct/indirect immunofluorescence. ELISA, enzyme-linked immunosorbent assay. IgA/IgG, immunoglobulin A/G. Dsg, desmoglein. Dsc, desmocollin. ^1^ IIF is commonly performed on monkey esophagus. The urothelium of monkey and rat bladder serve as excellent substrates for detecting anti-plakin autoantibodies to distinguish PNP from other pemphigus diseases.

**Table 2 jcm-11-03528-t002:** Clinical characteristics, target autoantigens and diagnosis of the subepithelial autoimmune bullous skin disorders (AIBDs).

Diseases	Clinical Features	Autoantigens [3]	Diagnosis [3,4]
Bullous pemphigoid (BP)	Tense blisters, erosions, and urticarial erythema on the trunk and flexural sites with preceding severe pruritus Uncommon mucosal involvement (10–20%) [14]	BP180, BP230	H&E: Subepidermal blisters with eosinophils and eosinophilic spongiosis DIF: Linear deposition of IgG and/or C3 on the BMZ IIF: Linear deposition of IgG on the BMZ; antibodies on the blister roof (salt-split skin [15]) ^1^ ELISA: Anti-BP180/Anti-BP230 antibodies
Mucous membrane pemphigoid (MMP)	Blistering occur predominantly on the oral cavity and conjunctiva which frequently healed with scarring [16,17,18]	BP180, BP230, laminin 332, α6β4 integrin	H&E: Similar to BP but with fewer eosinophils DIF: Linear deposition of IgG, IgA and/or C3 on the BMZ IIF: Linear deposition of IgG on the BMZ; antibodies on the blister roof and/or floor (salt-split skin) ELISA: Anti-BP180/Anti-BP230 and/or anti-laminin 332 antibodies
Linear IgA bullous dermatosis	Tense blisters located on the skin with involvement of the oral cavity (50%) String-of-pearls sign seen especially in the pediatric groups [19]	BP180 (LAD-1), type VII collagen	H&E: Subepidermal blisters with neutrophilic infiltrates DIF: Linear deposition of IgA on the BMZ IIF: Linear deposition of IgA on the BMZ; antibodies on the blister roof (salt-split skin) ELISA: Anti-BP180/Anti-LAD-1 IgA antibodies
Epidermolysis bullosa acquisita (EBA)	Tense bullae localized at the extensor aspects of the skin Nail dystrophy and esophageal stenosis may take place [20]	Type VII collagen	H&E: Subepidermal blisters with mixed infiltrates DIF: Linear deposition of IgG (less often IgM and IgA) and/or C3 on the BMZ IIF: Linear deposition of IgG on the BMZ; antibodies on the blister floor (salt-split skin) ELISA: Anti-type VII collagen antibodies
Dermatitis herpetiformis (DH)	Symmetrically distributed eruption of prurigo and tense vesicles on the skin Associated with celiac disease, a gluten-sensitive enteropathy	Epidermal/Tissue transglutaminase, endomysium, deamidated gliadin	H&E: Subepidermal blisters with papillary neutrophilic microabscess and scattered eosinophils DIF: Granular deposits of IgA on the dermal papillae IIF: Deposition of anti-endomysium IgA ELISA: Anti-epidermal/tissue transglutaminase antibodies, anti-deamidated gliadin IgA/IgG autoantibodies
Laminin γ1 pemphigoid	Tense bullae with urticarial erythema similar to BP [21,22] Some may associated with development of scars/milia (15.7%) [22]	p200 protein, laminin γ1	H&E: Subepidermal blisters with neutrophils and eosinophils infiltrates; some with papillary microabscess DIF: Linear deposition of IgG and/or C3 on the BMZ IIF: Linear deposition of IgG on the BMZ; antibodies on the blister floor (salt-split skin) ELISA: Anti-p200/Anti-laminin γ1 antibodies

Abbreviations: H&E, hematoxylin and eosin stain. DIF/IIF, direct/indirect immunofluorescence. ELISA, enzyme-linked immunosorbent assay. IgA/IgG, immunoglobulin A/G. BMZ, basement membrane zone. LAD-1, the linear IgA bullous dermatosis autoantigen. ^1^ IIF is commonly performed on monkey esophagus. Salt-split skin test separating the skin from the level above lamina lucida by immersing the specimen in 1 M sodium chloride solution elucidates the location of autoantibodies in subepidermal AIBDs.

**Table 3 jcm-11-03528-t003:** First-line therapeutic approaches to autoimmune bullous skin disorders (AIBDs).

Disease	First-Line Treatment
Intraepithelial AIBDs	
Pemphigus vulgaris	High dose systemic glucocorticoids (0.5 to 1.5 mg/kg/day prednisolone) and rituximab (either the lymphoma (weekly dosage of 375 mg/m^2^ for four consecutive weeks) or the rheumatoid arthritis (2 doses of 1000 mg separated by 2 weeks; may be repeated 6 months later) protocol)
	First-line adjuvants: Azathioprine (1 to 3 mg/kg/day), MMF (2 g/day) or mycophenolic acid (1440 mg/day)
Pemphigus foliaceus	The same as PV but usually with lower dosage
Pemphigus erythematosus	Systemic glucocorticoids (0.5 to 1 mg/kg/day prednisolone) and dapsone (100 to 200 mg/day)
IgA pemphigus	Systemic glucocorticoids (0.5 to 1 mg/kg/day prednisolone) and dapsone (100 to 300 mg/day)
Paraneoplastic pemphigus	Control of the underlying malignancy, systemic prednisolone (0.5 to 1 mg/kg/day) and rituximab (either the lymphoma (weekly dosage of 375 mg/m^2^ for four consecutive weeks) or the rheumatoid arthritis (2 doses of 1000 mg separated by 2 weeks; may be repeated 6 months later) protocol)
**Subepithelial AIBDs**	
Bullous pemphigoid	Systemic glucocorticoids (0.5 to 0.75 mg/kg/day of prednisolone) or high potency topical glucocorticoids, and tetracyclines ± niacinamide
Mucous membrane pemphigoid	Systemic glucocorticoids (0.25 to 0.5 mg/kg/day prednisolone) and dapsone (50 to 200 mg/day)
Linear IgA bullous dermatosis	Topical high potency glucocorticoids and dapsone (~2 mg/kg/day for children; 100 to 200 mg/day for adults)
Epidermolysis bullosa acquisita	High dose systemic glucocorticoids (1 to 1.5 mg/kg/day), dapsone (25 to 100 mg/day) and colchicine (0.6 to 1.2 mg/day)
Dermatitis herpetiformis	Gluten-free diet and dapsone (50 to 150 mg/day)

Abbreviations: IgA, immunoglobulin A. MMF, mycophenolate mofetil.

## Data Availability

Not applicable.

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
