# Peer review of "Current and Innovated Managements for Autoimmune Bullous Skin Disorders: An Overview"

_jcm, 2022, doi:10.3390/jcm11123528_

Round 1

Reviewer 1 Report

This is a well written article.  There is repetition of information that has been well known.  The initial part of the article describing the diseases clinical and histopathology can be eliminated since the focus is on treatment.  The current conventional treatment is described in a table and hence the text information RE this is repetition and can likely be eliminated.  The authors have done well to try to summarize current emerging treatments.  The text should be more focused on the mechanisms of action and logic behind the usage of medications of the trials. By deleting certain sections since they are not necessary or can be presented in a table, the authors can focus more on the mechanisms of action of the emerging medications and current outcomes/phase of studies.  

Reviewer 2 Report

Table 1 : Rituxan doses are missing.

2.1.1.2. Other Adjuvant Treatments: Why are Cytoxan, Dapson, IVIG called "adjuvant treatments". I suggest to find a more appropriate name

Section 2.1.1.3.2. is titled  Chimeric autoantibody receptor T-cells (CAAR-T-cells), but includes not only CAAR-T , but also Btk inh, BAFF inh, FcRn... The Btk inh, BAFF inh, FcRn... need to come under a different subtitle

Reviewer 3 Report

The authors provide an overview on possible and emerging treatments in AIBDs. the review is interesting but should needs some revision in the structure and in the wording. I liked the part on disease specific treatments, while the first part on epidemiology and diagnosis is too superficial. I recommend the consultation of an english writer to make this review interesting for the readership of the journal. Literature also lacks of certain works, i addedd some PMID as an example. Authors can also to add other works from the topic.

1.1.1 desmogleins are just one of the target antigens. Especially in PNP, which is mentioned in the paragraph other autoantigens play a major role. Please revise it

Line 55: PNP is also commonly associated with thymoma, please mention it (PMID 31293579)

1.1.2 here the authors should mention laminin gamma 1 pemphigoid (31695695, 19196964)

1.2 the epidemiology is way too general. What about PV and PF, which has peculiar geographical epidemiological differences? Epidemiology, if mentioned, should encompass all mentioned disorders (33228340)

1.4 Diagnosis should also discuss monkey oesophagus and rat bladder for pemphigus and PNP respectively . please specify how salt split skin works and which diseases show epidermal deposition and which ones present dermal deposition

2: Rituximab is approved only for pemphigus. The first sentence should therefore be revised.

2.1.1.3.1 please change the word miserably 

Efgartigimob is used also in immune thrombocytopenia, please correct

2.2.1.1 for my clinical experience, Doxyciclin does work very limitedly in BP - only in cases with localized involvement. I would at least mention that prednisone treatment is superior. 

Round 2

Reviewer 3 Report

none